# Robotic Rectal Resection for Rectal Cancer in Elderly Patients: A Systematic Review and Meta-Analysis

**DOI:** 10.3390/jcm12165331

**Published:** 2023-08-16

**Authors:** Rossella Reddavid, Silvia Sofia, Lucia Puca, Jacopo Moro, Simona Ceraolo, Rosa Jimenez-Rodriguez, Maurizio Degiuli

**Affiliations:** 1University of Turin, Department of Oncology, Division of Surgical Oncology and Digestive Surgery, San Luigi University Hospital, 10043 Turin, Italy; sofiaasilvia@gmail.com (S.S.); lucia.pucamed@gmail.com (L.P.); jacopo.moro@unito.it (J.M.); 2Nursing Degree Program, Department of Clinical and Biological Sciences, University of Turin, 10124 Torino, Italy; simona.ceraolo@unito.it; 3Department of Surgery, Hospital Universitario Virgen del Rocío, 41013 Sevilla, Spain; ros_j_r@hotmail.com

**Keywords:** rectal cancer, robotic surgery, elderly patients, survival outcomes, postoperative mortality, postoperative morbidity

## Abstract

Rectal cancer is estimated to increase due to an expanding aging population, thus affecting elderly patients more frequently. The optimal surgical treatment for this type of patient remains controversial because they are often excluded from or underrepresented in trials. This meta-analysis aimed to evaluate the feasibility and the safety of robotic surgery in elderly patients (>70 years old) undergoing curative treatment for rectal cancer. Studies comparing elderly (E) and young (Y) patients submitted to robotic rectal resection were searched on PubMed, Embase, and the Cochrane Library. Data regarding surgical oncologic quality, post-operative, and survival outcomes were extracted. Overall, 322 patients underwent robotic resection (81 in the E group and 241 in the Y group) for rectal cancer. No differences between the two groups were found regarding distal margins and the number of nodes yielded (12.70 in the E group vs. 14.02 in the Y group, *p* = 0.16). No differences were found in conversion rate, postoperative morbidity, mortality, and length of stay. Survival outcomes were only reported in one study. The results of this study suggest that elderly patients can be submitted to robotic resection for rectal cancer with the same oncologic surgical quality offered to young patients, without increasing postoperative mortality and morbidity.

## 1. Introduction

Rectal cancer (RC) in Europe accounts for 35% of colorectal tumors with approximately 125,000 cases per year, and the mortality rate is 4–10/100,000 cases annually. The median age at diagnosis is 70 , but it is estimated to increase in the near future due to population aging [1]. The risk of RC increases simultaneously with the elderly becoming highest among patients of about 85 years old. The population rate of octogenarians and nonagenarians is growing rapidly and at present patients ≥80 years old represent up to 24% of all patients submitted to colorectal surgical resection for cancer [2].

The optimal management of RC depends on the location and stage of the tumor. For early RC, both with mucosal or submucosal invasion, local excision could represent a curative treatment when technically feasible [3,4,5]. On the other hand, the standard of care for locally advanced RC is the trimodality approach with neoadjuvant chemotherapy and/or radiation therapy, surgical resection, and adjuvant chemotherapy [1,6].

Optimal surgical resection for RC must adhere to two main principles: removal of primary cancer with adequate circumferential and distal margins and performance of a total mesorectal excision (TME) to remove all draining lymphatics [7,8,9]. The quality of rectal surgery is based upon three indicators: TME quality, negative circumferential and distal margins, and number of lymph nodes yielded [10,11].

Anterior resection for RC can be performed with three different approaches: open, laparoscopic, and robotic. Recently, robotic surgery has been introduced to overcome the limitations of laparoscopy. Robotic surgery takes advantage of three-dimensional vision, instruments with articulating motions, and platform stability with tremor elimination [12]. These technical improvements could potentially further the two main objectives of cancer treatment, i.e., survival outcomes and enhanced quality of life.

Consistent with the existing literature, robotic surgery seems to improve short-term outcomes (lower conversion rates, shorter length of stay, and improved 30-day overall survival rate) when compared with laparoscopy, whereas survival outcomes remain the same [13].

Indeed, the faster postoperative recovery achieved by robotic surgery is very valuable for elderly patients leading to a quicker return to an active life.

To the best of our knowledge, the majority of the clinical data used to guide treatment decisions originated from trials that enrolled patients aged less than 75. Elderly patients are rarely included in trials, representing less than 10% [14]. Therefore, they are treated following guidelines based upon trials involving young patients, with the main risk being undertreatment or overtreatment. Several authors identified 70 as a reasonable cutoff age for defining elderly patients [15,16]. The aim of this study is to evaluate the available evidence of the benefits provided by robotic surgery in elderly patients (>70 years old) undergoing curative treatment for RC in terms of short- and long-term postoperative outcomes.

## 2. Materials and Methods

The present systematic review was performed in accordance with the Preferred Reporting Items for Systematic Reviews and Meta-analysis (PRISMA) guidelines [17] and it was registered with the International Prospective Register of Systematic Reviews (PROSPERO) with the ID number CRD42023433816.

### 2.1. Literature Search and Selection

The literature search was conducted to identify all studies focusing on robotic resection for RC in elderly patients (>70 years old). No limitations of language, publication date or status were adopted. On 20 February 2023, with the support of the University of Turin Library ‘Biblioteca Federata di Medicina‘ (Federated Library of Medicine), we systematically searched all databases including PubMed, Embase, and the Cochrane Library. The search sequences utilized for each database are reported in Appendix A. The research query was launched again on 31 May 2023 to check for studies published in the meanwhile.

The primary endpoints were 30- and 90-day postoperative mortality rates. The secondary outcomes were the length of stay, intraoperative complications, operative mortality, postoperative complications, 5-year overall survival (OS), disease-free survival (DFS), and recurrence rates.

All the included papers had to comply with the following PICO items. (i) Patient: elderly patients undergoing curative treatment for rectal cancer; (ii) intervention: rectal cancer resection performed with a robot-assisted approach; (iii) comparison: Another surgical approach to rectal resection; (iv) outcomes: data regarding survival rates and postoperative complications.

Inclusion criteria were proven diagnosis of rectal cancer, elderly patients (>70 years old), robotic surgery, articles with at least one outcome of interest, English articles and papers published in the last 10 years.

Exclusion criteria were conference papers, oral communications, abstracts, letters, editorials, reviews, meta-analyses, duplicated publications, and papers that could not be located for a multitude of reasons (not retrieved).

Four of the authors (J.M., L.P., R.R. and S.S.) independently screened the articles, firstly, by titles, subsequently, by abstracts, and, finally, by full texts. Disagreements were resolved by team discussion. The authors of one study were contacted with the aim of including data according to the pre-established cutoff age of 70.

### 2.2. Statistical Analysis

For categorical outcomes, we performed pooled analyses using the Mantel–Haenszel method for odds ratios (ORs) with 95% confidence intervals (CI) with fixed effects. The DerSimonian and Laird method was used for random effects. Results from pooled analyses of continuous variables are reported as mean differences with 95% CI. We calculated risk differences when some studies reported no events. When selected studies reported continuous variables as ‘median’ and ‘IQR’ or ‘min/max’ range, we used a validated method to estimate ‘mean’ and ‘SD’ and pool analyses.

### 2.3. Heterogeneity

The degree of heterogeneity between studies was assessed using the chi-square (χ^2^) test and the I^2^ statistic, where values less than 40% suggest good homogeneity for the reliability of the meta-analysis. In the case of a relevant heterogeneity level, a random effect model was used.

### 2.4. Quality Assessment

The methodological quality of the included studies was assessed using the Newcastle–Ottawa Scale (NOS) [18]. Two different researchers (SS and PL) assessed quality independently. A score was assigned for methods of selection (such as case definition, representativeness of cases, and selection of controls), comparability of cases and controls on the basis of the design and analysis, and exposure (such as ascertainment of exposure, the same method of ascertainment for cases and controls, and non-response rate).

The studies with a score <6 were excluded from the present meta-analysis due to their low quality. Otherwise, all the papers that scored ≥6 were included because they were considered to be of good–high quality.

All analyses were performed using the software RevMan version 5.0 and a two-sided *p* value less than 0.05 was considered statistically significant.

## 3. Results

### 3.1. Literature Selection

The selection of the literature and studies is reported in Figure 1 according to PRISMA guidelines [17]. The search resulted in 2501 articles (PubMed 817, Cochrane CENTRAL 89, Embase 1595) with the assistance of the ‘Biblioteca Federata di Medicina’ (Federated Library of Medicine). After the deduplication process, 691 papers were removed. Thereafter, the articles were screened by title which resulted in the exclusion of a further 1774 irrelevant papers. The remaining 36 articles were first selected by abstract and, subsequently, by full-text examination. Finally, only three papers meeting the inclusion criteria were chosen for the present systematic review [19,20,21].

### 3.2. Study and Patient Characteristics

All the included papers were retrospective studies with a small sample size comparing elderly patients (E group) with young patients (Y group). The total number of patients was 322 (ranging from 15 to 156 per trial) from three countries (Spain, Taiwan, and Italy), with 81 patients in the E group and 241 patients in the Y group. The mean age was 76.54 in the E group and 58.19 in the Y group, both groups having approximately 60% male participants.

The majority of the elderly patients had an ASA III score (69%), while most of the young patients had an ASA II score (71%).

The rate of neoadjuvant treatment was 85.1% in the E group and 59% in the Y group. The characteristics of the studies are reported in Table 1.

### 3.3. Risk of Bias

The quality assessment of the included studies is reported in Table 2. The overall score of each paper was 8, which means that all the included studies in our meta-analysis were of good quality.

### 3.4. Oncologic Surgical Quality

The data regarding the oncologic surgical quality are detailed in Table 3. All studies reported the distance from the distal margin. No significant difference was observed in the distal margins between the two groups [Z = 0.64, MD = −0.47, 95% CI (−1.92, 0.97), *p* = 0.52] with a heterogeneity of 62% among the studies (I^2^ = 62%, *p* = 0.07); therefore, a random effects model was used. The forest plot of distal margins is reported in the Appendix A. All studies described the number of lymph nodes yielded (12.7 ± 2.31 in the E group vs. 14.02 ± 1.83 in the Y group). No differences resulted between the groups [Z = 1.41, MD = −1.51, 95% CI (−3.62, 0.59), *p* = 0.16]; no heterogeneity among the studies (I^2^ = 0%, *p* = 0.96) (Figure 2) justified the use of a fixed effects model.

### 3.5. Postoperative Outcomes

Postoperative outcomes are detailed in Table 4. All studies reported the conversion rate, which was higher in the E group (8.6%) as compared with the Y group (5.8%), without a significant difference in the pooled analysis [Z = 0.48, RD = 0.02 95% CI (−0.05, 0.09), *p* = 0.63]. No heterogeneity resulted from the analysis. For the analysis of the conversion rate, the risk difference was used as the measure of the effect because it can be estimated for any study, even in the case of no events in either group selected in the study. The forest plot of conversion rate and complications are reported in the Appendix A.

The length of stay was detailed by all authors and it was similar between the two groups (16.29 ± 4.08 vs. 13.19 ± 1.56) [Z = 0.58, *p* = 0.56]. The heterogeneity was 22% (Figure 3).

The overall complications were reported by all the included studies, and a little difference was found between the groups (20.9% in the E group vs. 23.2% in the Y group).

The primary outcomes of the present meta-analysis were the 30- and 90-day postoperative mortality rates. All articles reported a 1-month mortality rate, while none reported a 3-month mortality rate. No postoperative mortality occurred; consequently, the pooled analysis was similar without significant differences (*p* = 1.00) [Z = 0.00, RD = 0.00, 95% CI (−0.03, 0.03)]. No heterogeneity was found among the studies (I2 = 0%, *p* = 1.00) (Figure 4); therefore, a fixed effects model was used. Regarding the postoperative mortality rate outcome the risk difference was calculated as an effect measure because of the absence of cases in all the study groups.

### 3.6. Survival Outcomes

Only Su [21] reported detailed survival outcomes (DFS, OS, local and distant recurrence); therefore, the meta-analysis was not applicable. The authors reported a 5-year DFS of 78.6% in the E group and 68.5% in the Y group without a significant difference (*p* = 0.719). The 5-year OS was similar between the two groups (82.4% and 88.3%, respectively, *p* = 0.390), as well as the local and distant recurrence parenthesis (0% E group vs. 9.5% Y group, *p* = 0.168 and 6.7% E group vs. 17.5 Y group, *p* = 0.234, respectively).

## 4. Discussion

This systemic review and meta-analysis have found new evidence based on the most recent literature. No differences between the two groups were found regarding distal and proximal margins and the number of nodes yielded (12.70 in the E group vs. 14.02 in the Y group, *p* = 0.16). The conversion rate, postoperative morbidity, mortality, and length of stay were similar in the groups.

Several studies [22,23] have reported decreased survival rates in elderly patients with RC (>65 years old) compared with young patients, which is mainly due to inappropriate care characterized by either undertreatment due to their advanced age or overtreatment due to their degree of frailty [24].

Furthermore, worldwide guidelines for elderly patients are not accurate because this population is often excluded or less represented in trials. The International Society of Geriatric Oncology (SIOG) has developed guidelines for the management of colorectal cancer (CRC) in this age group with the aim of reducing discrepancies in the treatment of young and elderly patients. SIOG recommendations state that “age is not a contraindication to MIS, while data seem to point to laparoscopy as the preferred option to perform TME surgery. This is particularly relevant for elderly patients for whom a faster return to an active life, achieved with a less invasive but equally oncologically appropriate procedure, is the most desirable outcome” [25]. Robotic surgery was initially introduced with the aim of reducing conversion rates but no RCTs show any advantages in performing robotic surgery. However, this minimally invasive approach can theoretically reduce surgery stress, thus leading to a faster recovery. As a result, robotic surgery can be a valid option for elderly patients. However, the only two RCTs [26,27] comparing robotic vs. laparoscopic approaches for patients with RC have little representation of elderly patients with a median age lower than 65.

The ROLARR trial [26] enrolled 471 patients with RC suitable for resection with the aim of comparing robotic versus laparoscopic surgery for conversion risk. This trial failed to demonstrate the superiority of the robotic approach in terms of reduction of conversion rates compared with laparoscopic surgery. The REAL study [27] enrolled 1240 patients eligible for resection for RC with the aim of comparing the robotic versus laparoscopic approach in terms of surgical quality and long-term oncological outcomes. The primary outcome was a 3-year local recurrence, but it was not evaluated because the follow-up was not concluded at that time. Secondary short-term outcomes showed the superiority of robotic surgery compared with the laparoscopic approach in terms of higher rates of macroscopic complete resection, shorter length of stay, reduction in abdominoperineal resections, lower conversion rates, reduction in blood loss and lower intra-operative complications rate. Minor single center RCTs comparing robotic surgery to laparoscopy for RC are present in the literature but the quality of evidence is very low due to the small sample size. Recently, a systematic review and meta-analysis has investigated the safety and efficacy of robotic versus laparoscopic surgery for RC [13]. The review analyzed 41 studies (retrospective, prospective and randomized controlled studies) with 19,731 patients with a median age of 62. The authors have concluded that robotic surgery provides many advantages compared with the laparoscopic approach in terms of shorter operation time, lower conversion rates, shorter length of stay and improved 30-day overall survival rate.

To date, there have only been three retrospective studies comparing elderly vs. young patients submitted to robotic resection for RC. The median age of the elderly group was 76.54, which was significantly higher than the median age reported in the main studies on robotic surgery for RC [13,26,27].

The oncological surgical quality was similarly achieved in both groups in terms of distal and proximal margins and the number of lymph nodes yielded. The definition of “optimal” distal margin (DM) in RC surgery is still being debated. Initially, the DM was established as at least 5 cm [28] but the introduction of both TME and neoadjuvant chemoradiotherapy has challenged this concept and 2 cm is now considered to be safe [29]. Recently, several studies have shown that a DM of less than 1 cm is sufficient in patients with middle and low RC submitted to neoadjuvant treatment [30]. The results of this study are in line with these recommendations.

A milestone In oncologic surgery is the lymph node count, which is essential to ensure lymph node examination and, consequently, an accurate staging. Current guidelines require a minimum of 12 nodes to define good node staging [1,6]. However, the optimal minimum number of lymph nodes retrieved in patients with RC submitted to neoadjuvant therapy is still under evaluation [31,32]. In this study, the mean of nodes yielded in young patients is reported to be higher than 12, while in the elderly population, it is 12.7 ± 2.31 (*p* = 0.16). A large retrospective study reported a wide variation in lymph node yield associated with patient age (patients younger than 50 received an adequate node evaluation twice as high as those aged 71 or older) [33]. Furthermore, Mekenkamp demonstrated that patients aged >60 are related to significantly lower lymph node retrieval [34].

A meta-analysis investigating the feasibility and safety of robotic colorectal surgery reported an advantage of robotic surgery in terms of lower conversion rate, lower intra-operative blood loss, and shorter length of stay compared with the laparoscopic approach. However, no subgroup analysis was conducted for the elderly group.

In the present meta-analysis, no remarkable differences resulted in terms of operative and postoperative outcomes between young and elderly patients. The rate of conversion is higher in the E group (8.6%) without significant differences compared with the Y group (5.8%). The higher percentage of conversion in elderly patients is probably related to the high rate of advanced T stage. Several studies reported a strict relationship between larger and locally advanced tumors and a higher percentage of conversion [35,36]. Furthermore, in this study, the conversion rates are consistent with those reported in the ROLARR trial [26].

A retrospective study with a large sample size reported a significantly longer length of stay in elderly patients compared with younger patients submitted to colorectal surgery (24 vs. 20 days, *p* = 0.002). The authors attributed this result to the frequent monitoring of elderly patients in the Intensive Care Unit [37]. However, no differences were observed in this meta-analysis.

Consistent with the literature [38], age is an independent risk factor for increased rates of postoperative complications and mortality; nevertheless, a great deal of evidence supports that the real risk factor is the degree of frailty rather than the mere age [39,40,41].

This meta-analysis confirms that age is not a risk factor for worse postoperative outcomes; indeed, no differences are present between young and elderly patients in terms of postoperative mortality and morbidity.

This study has some limitations. Firstly, the number of patients included is small, which reduces the statistical power of this meta-analysis. Secondly, to date, no RCTs regarding this topic are present in the literature and the included papers are retrospective studies. Finally, the included studies miss a number of data, with only one study reporting survival outcomes.

## 5. Conclusions

The present study is the largest analysis investigating the role of robotic surgery for elderly patients with rectal cancer. The results suggest that elderly patients (>70 years old) can be submitted to robotic resection for RC with the same oncologic surgical quality offered to young patients, without increasing postoperative mortality and morbidity. Future high-quality RCTs are needed to confirm that robotic surgery for rectal cancer resection in the elderly is feasible and safe.

## Figures and Tables

**Figure 1 jcm-12-05331-f001:**
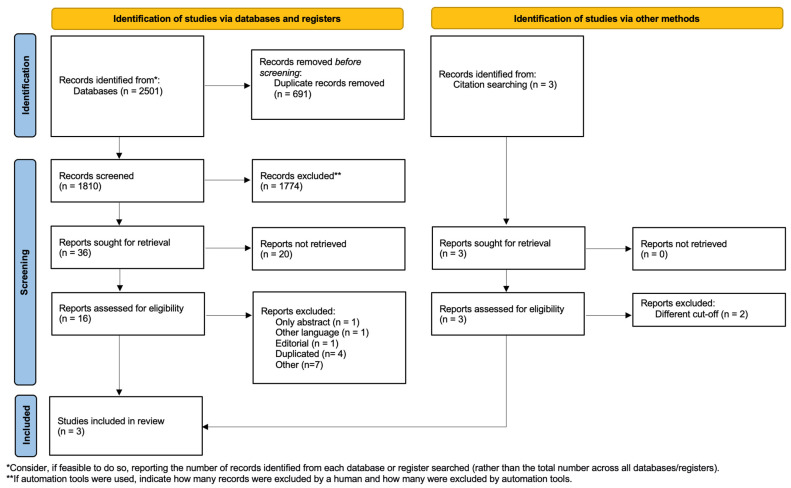
PRISMA 2020 flow diagram for new systematic reviews which included searches of databases, registers, and other sources.

**Figure 2 jcm-12-05331-f002:**
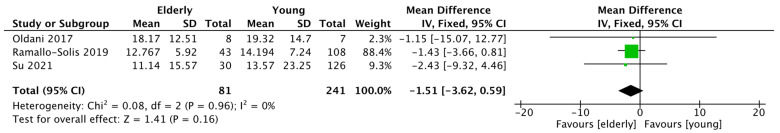
Harvested lymph nodes [19,20,21].

**Figure 3 jcm-12-05331-f003:**
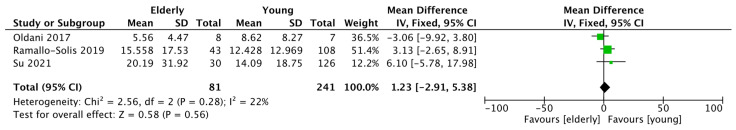
Length of stay [19,20,21].

**Figure 4 jcm-12-05331-f004:**
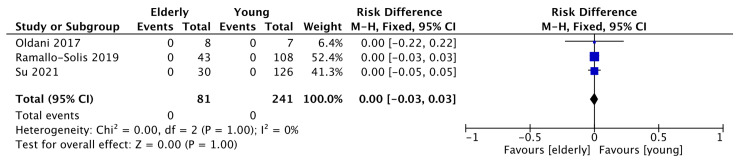
Thirty-day postoperative mortality [19,20,21].

**Table 1 jcm-12-05331-t001:** Features of selected studies.

Study	Country	Study Design	Patients	Mean Age (SD)	Sex (M/F)	ASA (I/II/III/IV)	Neoadjuvant Therapy Yes/No	cT Stage (T1/T2/T3/T4)	cN Stage (N0/N1/N2)	cM Stage (M0/M1)
Oldani [19] 2017	Italy	Retrospective study	7 young	57.12 (5.5)	4M/3F	1/6/0/0	4/3	ND	ND	6/1
			8 elderly	81.9 (6.28)	4M/4F	0/2/6/0	2/6	ND	ND	7/1
Ramallo-Solis [20] 2019	Spain	Retrospective analysis of a prospective database	108 young	60 (7.69)	65M/43F	12/69/27/0	71/37	4/25/78/1	41/41/26	102/6
			43 elderly	74.74 (3.566)	29M/14F	0/15/28/0	24/19	1/11/30/1	14/20/9	43/0
Su [21] 2021	Taiwan	Retrospective study	126 young	56.7 (7.9)	77M/49F	0/96/30/0	90/80	4/26/81/15	55/45/26	ND
			30 elderly	77.7 (5.63)	17M/13F	0/7/22/1	22/8	¼/23/2	13/16/1	ND
**Tot.**			**241 young**	**58,19 (7.11)**	**144M/95F**	**13/171/57/0**	**143/111**			
			**81 elderly**	**76,54 (5.29)**	**50M/31F**	**0/24/56/1**	**69/43**			

SD: standard deviation; M: male; F: female; ASA: American Society of Anesthesiologists; cT: clinical Tumor stage; cN: clinical lymph mode stage; cM: clinical metastases stage; T1–T4: tumor depth defined according to the criteria of the AJCC/International Union Against Cancer (UICC); N0–N2: presence of any lymph node metastases defined according to the criteria of the AJCC/UICC; M0–M1: presence of any distal metastases defined according to the criteria of the AJCC/UICC; ND: not defined.

**Table 2 jcm-12-05331-t002:** Newcastle-Ottawa Scale (NOS). Every point of the NOS is represented by a star in the table (*, **).

Study	Representativeness of Exposed Cohort	Selection of Non-Exposed Cohort	Ascertainment of Exposure	Absence of Outcome of Interest at Start of Study	Comparability of Cohorts on the Basis of Design or Analysis	Assessment of Outcome	Follow-Up Enough for Outcome to Occur	Adequacy of Follow-Up of Cohorts	Score	Quality
Oldani [19]	*****	*****	*****	*****	******	*****	*****	-	8	Good
Ramallo-Solis [20]	*****	*****	*****	*****	******	*****	*****	-	8	Good
Su [21]	*****	*****	*****	*****	******	*****	*****	-	8	Good

**Table 3 jcm-12-05331-t003:** Pooled analysis of oncological outcomes.

Outcome	N of Studies	Means (SD)		Heterogeneity of Trials		*p*-Value for Differences across Studies	Mean Difference (C.I. 95%)
				*p*-Value	I^2^ Statistic		
Distal margin	3	Young 2.93 (0.46)	Elderly 2.37 (0.78)	0.07	62%	0.52	−0.47 (−1.92, 0.97)
N. of harvested lymph nodes	3	Young 14.02 (1.83)	Elderly 12.70 (2.31)	0.96	0%	0.16	−1.51 (−3.62, 0.59)

N: number, SD: standard deviation, C.I.: confidence interval.

**Table 4 jcm-12-05331-t004:** Pooled analysis of intraoperative and postoperative outcomes.

Outcome	No. of Studies	No. of Events or Mean (SD)		Heterogeneity of Trials		*p*-Value for Differences across Studies	OR, Mean or Risk Difference (C.I. 95%)
				*p*-Value	I^2^ Statistic		
Conversion rate	3	Young 14/241	Elderly 7/81	0.73	0%	0.63	0.02 (−0.05, 0.09)
Length of stay (LOS)	3	Young 13.19 (1.56)	Elderly 16.29 (4.08)	0.28	22%	0.56	1.23 (−2.91, 5.38)
Overall complications	3	Young 56/241	Elderly 17/81	0.68	0%	0.71	0.89 (0.48, 1.65)
Postoperative mortality	3	Young 0/241	Elderly 0/81	1.00	0%	1.00	0.00 (−0.03, 0.03)

No: number; SD: standard deviation; OR: odds ratio; C.I.: confidence interval.

## Data Availability

Data are available upon request.

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
