# Peer review of "Robotic Rectal Resection for Rectal Cancer in Elderly Patients: A Systematic Review and Meta-Analysis"

_jcm, 2023, doi:10.3390/jcm12165331_

Round 1

Reviewer 1 Report

The authors conducted a meta-analysis comparing outcomes of robotic rectal surgery between young end elderly patients.

The topic indeed seems a less studied field, however the authors' presentation of the results leaves me with some doubts.

1.) It is unclear to me how did the authors select 70 y.o. as the age dividing the groups for comparison. This should be specified since in discussion the authors rightly recognised the degree of frailty as the most important risk factor. The line of 'elderly' seems to be pushed forward, therefore explanation of authors' decision should be given.

2.) One of the articles included in the analysis - 'Infuence of robotics in surgical complication rate in elderly population with rectal cancer' by Ramallo‑Solis et al - divided patients into groups according to a different age pattern: under 66 year old, between 66 and 79 year old and 80 year old and above. How did authors retrieve the specific groups of patients younger and older than 70 y.o. from this study? 

Apart from the above, I would give the authors small remarks:

- I think the first paragraph of the discussion is more suitable for introduction.

- Instead of the mentioned paragraph in the beginning of discussion I recommend summary of the results. 

- In my opinion the conclusion in the last paragraph should be a separate section headlined 'Conclusions'. 

- PRISMA chart should an image be of higher quality - currently it is blurred.

The authors endevoured an analysis in an interesting topic, however the overall quality of provided evidence (partially due to the small number of studies and patients) is moderate. 

Author Response

  • It is unclear to me how did the authors select 70 y.o. as the age dividing the groups for comparison. This should be specified since in discussion the authors rightly recognised the degree of frailty as the most important risk factor. The line of 'elderly' seems to be pushed forward, therefore explanation of authors' decision should be given

Reply.

Thank you for your comment. We selected the cut-off of 70 years old because several authors showed that this age is associated with increased colorectal cancer-specific death and poor colorectal cancer-specific survival (Fu et Al  PeerJ 2019, Podda et al. World Journal of Emergency Surgery 2021).

We have added these comments at page 4, in the “Introduction” section.

  • One of the articles included in the analysis - 'Infuence of robotics in surgical complication rate in elderly population with rectal cancer' by Ramallo‑Solis et al - divided patients into groups according to a different age pattern: under 66 year old, between 66 and 79 year old and 80 year old and above. How did authors retrieve the specific groups of patients younger and older than 70 y.o. from this study? 

Reply.

Thank you for your comment, which is fundamental for improving the results of our work.

We proceeded to contact the authors of this included article and we asked them their database. Then we redid all the analysis.

We have added these new results at page 3, in the “Results” section.

3) I think the first paragraph of the discussion is more suitable for introduction.

Reply.

Thank you for your comment. We proceeded to modify the discussion and the introduction sections in agreement with your suggestions.

We moved this paragraph at page 2 in the “Introduction” section.

4) Instead of the mentioned paragraph in the beginning of discussion I recommend summary of the results. 

Reply.

Thank you for your comment. We introduced a summary of the results in the discussion section in agreement with your suggestions.

We have added this paragraph at page 11 in the “Discussion” section.

5) In my opinion the conclusion in the last paragraph should be a separate section headlined 'Conclusions'. 

Reply.

Thank you for your comment. We moved the conclusions in a separated section named “Conclusions”

6) PRISMA chart should an image be of higher quality - currently it is blurred.

 Reply.

Thank you for your comment. We modified the PRISMA image.

Reviewer 2 Report

Overall, the paper needs further refinement in terms of English use.

The introduction could potentially benefit from the following suggestions:

Please consider removing the first paragraph, as it is too general and does not serve the purpose of the paper.

- Additionally, please consider removing the history of laparoscopic rectal surgery from the sixth paragraph.

- This section includes ample debates about previously published papers regarding the target topic. I would recommend to move them in the Discussion section.

-  Moreover, when the authors wish to refer to a paper such a clinical trial for example, they should draw a conclusion on the study rather than quoting the exact paragraph.

        The Materials and methods section could be improved by:

-        Stating more clearly the inclusion and exclusion criteria.

-        Clarifying if all papers or just the ones using English language were included.

-    Explaining what the authors meant by ”not retrieved” papers.

The Results section could be improved by re-assessing the inclusion criteria, as 3 papers seem insufficient, without proper presentation of the primary and secondary proposed outcomes. The authors should consider including papers no older than 5 years, with a minimum number of 50-100 patients/study.

Finally, the Discussions abound of irrelevant details, some of them repeating the information provided by the introduction. 

To summarize, I believe that the aim of the study needs to be more clear and the search strategy should be structured according to the proposed aim.

I believe that further refinement of English language is needed, as there can be identified some grammar mistakes, as well as improper topic of sentences and discordance between the subject and corresponding verbs.

Author Response

  • Overall, the paper needs further refinement in terms of English use.

 Reply.

Thank you for your comment. We modified the text in agreement with an English teacher revision.

2) The introduction could potentially benefit from the following suggestions:

-  Please consider removing the first paragraph, as it is too general and does not serve the purpose of the paper. 

Reply.

Thank you for your comment. We removed the first paragraph of introduction.

3) The introduction could potentially benefit from the following suggestions:

- Additionally, please consider removing the history of laparoscopic rectal surgery from the sixth paragraph.

Reply.

Thank you for your comment. We removed the sixth paragraph of introduction.

4) The introduction could potentially benefit from the following suggestions

- This section includes ample debates about previously published papers regarding the target topic. I would recommend to move them in the Discussion section.

Reply.

Thank you for your comment. We moved that paragraph to the Discussion section.

5) The introduction could potentially benefit from the following suggestions:

-  Moreover, when the authors wish to refer to a paper such a clinical trial for example, they should draw a conclusion on the study rather than quoting the exact paragraph.

Reply.

Thank you for your comment. We modified the text in agreement with your suggestions.

        6) The Materials and methods section could be improved by:

-        Stating more clearly the inclusion and exclusion criteria. 

Reply.

Thank you for your comment. We modified the text in agreement with your suggestion.

We have added these sentences at page 4 in the “Methods” section.

  7) The Materials and methods section could be improved by:

-        Clarifying if all papers or just the ones using English language were included.

Reply.

Thank you for your comment. We modified the text in agreement with your suggestion.

We have added this sentence at page 4 in the “Methods” section.

8) The Materials and methods section could be improved by:

-    Explaining what the authors meant by ”not retrieved” papers.

Reply.

Thank you for your comment. We reported the PRISMA definition of “not retrieved” papers in the “Method” section.

We have added this sentence at page 4 in the “Methods” section.

9) The Results section could be improved by re-assessing the inclusion criteria, as 3 papers seem insufficient, without proper presentation of the primary and secondary proposed outcomes. The authors should consider including papers no older than 5 years, with a minimum number of 50-100 patients/study.

Reply.

Thank you for your comment. We modified the text in agreement with your suggestions.

We have added this sentence at page 4 in the “Methods” section.

10) the Discussions abound of irrelevant details, some of them repeating the information provided by the introduction. 

Reply.

Thank you for your comment. We modified the text in agreement with your suggestions.  

11)I believe that further refinement of English language is needed, as there can be identified some grammar mistakes, as well as improper topic of sentences and discordance between the subject and corresponding verbs.

Reply

Thank you for your comment. We modified the text in agreement with an English teacher revision.

Round 2

Reviewer 2 Report

I would like to congratulate the authors for the current form of the manuscript.

I think the paper is suitable for publication.